# Agreement between the SHAPES Questionnaire and a Multiple-Sensor Monitor in Assessing Physical Activity of Adolescents Using Categorial Approach: A Cross-Sectional Study

**DOI:** 10.3390/s21061986

**Published:** 2021-03-11

**Authors:** Ivan Radman, Maroje Sorić, Marjeta Mišigoj-Duraković

**Affiliations:** 1Faculty of Kinesiology, University of Zagreb, 10000 Zagreb, Croatia; maroje.soric@kif.unizg.hr (M.S.); marjeta.misigoj-durakovic@kif.unizg.hr (M.M.-D.); 2Faculty of Sport, University of Ljubljana, 1000 Ljubljana, Slovenia

**Keywords:** physical activity assessment, recall questionnaire, self-report, multiple-sensor monitor, accelerometery, kappa statistics

## Abstract

This study aimed to evaluate the agreement between a 7-day recall questionnaire and multiple-sensor monitor in identifying sufficiently active adolescents. A total of 282 students involved in the CRO-PALS study were randomly selected for a device-based measurement of physical activity (PA) using the SenseWear Armband device (SWA) no more than three weeks before or after having fulfilled the SHAPES questionnaire. Valid data was obtained from 150 participants (61 boys; 89 girls) and included in the analysis. In boys, SHAPES exhibited high specificity (92.3%), overall percent agreement (85.0%), and significant agreement (κ = 0.32, *p* = 0.014) with the SWA in recognising sufficiently active individuals. Conversely, no agreement was detected for quartiles of PA, although boys that were classified in the first and in the fourth quartile by SHAPES differed in device-based measured duration of MVPA (134 [95%CI: 109–160] vs. 87 [95%CI: 65–108], *p* = 0.032); and VPA (39 [95%CI: 23–56] vs. 14 [95%CI: 6–22], *p* = 0.011). In girls, no significant agreement between the two methods was found in any of the analyses. It appears that the SHAPES questionnaire is effective to identify individuals that comply with PA recommendations and to distinguish between the most active and the least active individuals for adolescent boys, but not for girls.

## 1. Introduction

Chronic non-communicable diseases (NCDs) such as cardiovascular disease, cancer, chronic respiratory diseases and diabetes are the leading causes of disability and mortality worldwide. Various health-related associations including the World Health Organization (WHO) account insufficient physical activity (PA) among the small set of behavioural factors contributing to development of NCDs. Moreover, insufficient PA has been recognized as a fourth leading risk factor for global mortality, responsible for nearly 6% of all-cause deaths worldwide [1]. However, there is an indisputable body of evidence suggesting that habitual PA reduces the risk for early death and is an effective prevention for at least 25 chronic medical conditions [2].

Since behavioural patterns developed during early life stages tend to remain during adulthood [3,4], special attention has been given to promote meeting the recommended amount of PA through the periods of childhood and adolescence. According to the global health strategy issued by WHO, a minimum of 60 min moderate-to-vigorous physical activity has been recommended to preserve and improve cardiovascular and metabolic health (WHO 2004). At the same time, available studies report on the increasing trend of physical inactivity among the population of adolescents worldwide [5,6].

Both global and national policies aimed at enhancing general levels of health among the adolescent population embrace the systematic monitoring of health-related habits and risk behaviours during the school period. In order to control physical activity among adolescents, a wide range of devices such as accelerometers, pedometers, multiple-sensor PA monitors, fitness trackers and smartphone technologies have been recommended as valid and accurate [7,8,9,10]. However, because of their limited availability and relatively high costs, most of the large-scale population studies rely on subjective methods such as PA questionnaires and recalls [11]. Relative to activity monitors, the concept of recalling seems to be advantageous in identifying a range of PAs, including cycling and water activities, and thus the overall PA.

The currently available literature suggests the limited ability of subjective methods to provide accurate data on either energy expenditure (EE) or time spent in physical activity [12,13]. Although authors recommend several questionnaires as suitable for public health research and practice, 7 day-recall questionnaires tend to overreport both above measures [14,15,16] and are generally considered to have slightly lower criterion-related validity in comparison to device-based methods [17].

It is common that raw measures of total EE and average time spent at different intensities of physical activity have a crucial role while designing individual health-enhancing programs. On the other hand, detecting the portions of individuals meeting health-related recommendations [18] or those belonging to a subgroup at greater risk of engaging in risk behaviours within specific population is one of the priorities in planning the public health strategies [19]. Consequently, one of the goals of PA assessment in observational epidemiological studies is to categorise individuals into quantiles of PA duration [20]. Indeed, because of both the previously mentioned issues and the discrepancy between self-report and device-based measurement of PA, it has not been recommended to analyse the outcome of recall instruments as continuous variables, but rather transformed into a form of rank or category [12,17,21]. Because of particularly variable individual levels of PA among children and adolescents, ranging from nearly inactive to active and athletic, categorising PA-duration seems to be clinically important in these populations. Although the current health guidelines promote frequent daily engagement in moderate and vigorous PA, even low regular participation in PA has been encouraged as beneficial for health status in adolescents [22,23,24]. Studies reporting the level of agreement between questionnaires and PA monitors for categorising adolescents as physically active or ranking them as lowly/highly active (i.e., under greater or lower risk of engaging in risk behaviour) might be advancing for public health practice. Hence, the purpose of the current study was to evaluate the level of agreement between subjective and device-based methods for habitual PA assessment in: (i) identifying insufficiently active adolescents according to the current PA guidelines and (ii) ranking individuals into quartiles of PA duration.

## 2. Materials and Methods

### 2.1. Study Participants

This study is a part of the CRO-PALS, an observational, longitudinal study designed to investigate the health-related habits and risk behaviour among the adolescents in the city of Zagreb (Croatia) during their 4-year-long high-school education. The sampling procedure and study design have been described thoroughly elsewhere [25,26]. In brief, to select a representative group of urban adolescents, a stratified two-phase random sampling method was applied. Within the first phase, all secondary schools in the city of Zagreb (*n* = 86) were stratified by type as “grammar”, “vocational” and “private”. According to the original proportion of corresponding school-types and average number of eligible students per school, 13 public (8 vocational and 5 grammar schools) and 1 private school (grammar school) were selected. Subsequently, the second phase encompassed randomizing half of all first-grade classes enrolled in each of the primary elected schools. Finally, 1408 students registered in selected classes were invited for participation, whereas 903 agreed to be involved in the study (response rate = 64%). All study procedures were conducted between March and June 2014. Prior to participating in in the study, all students and their parents signed an informed consent in accordance with the Declaration of Helsinki. The study was approved by the Institutional Review Board of the Faculty of Kinesiology, University of Zagreb, Croatia under No. 1009-2014.

### 2.2. Study Design

After having confirmed the participation in the study, a group of selected adolescents were invited to fulfil the 7-day recall PA questionnaire. Afterwards, from the above sample, 5 secondary schools counting 282 students were randomly selected for a device-based measurement of PA. They were asked to wear a multiple-sensor physical activity monitor for the quantification of PA for 5 consecutive days, no more than three weeks before or after having fulfilled the questionnaire. Out of the total number of students invited for subjective assessment and device-based measurement of PA, 189 students submitted valid results of both methods. Out of that number, 150 (61 boys and 89 girls) students reported that both the device-based-measurement period and the subjective-assessment period corresponded to their habitual/usual PA, which is intended to be observed in this study.

In order to accomplish the purpose of the study, a comparison of single categorial scores between the questionnaire and valid multiple sensor monitor was applied. The cross-sectional comparison design was applied to evaluate the construct validity, i.e., the ability of the questionnaire to accurately categorise adolescents versus multiple sensor monitoring in three dichotomous variables “sufficient activity” (based on the cut-off of 60 min of moderate-to-vigorous PA [MVPA] daily, which categorises participants as sufficiently active/insufficiently active), “low activity” (i.e., values fitting the 4th quartile of PA measured by each instrument, which categorises participants as last quartile/other quartiles), “high activity” (i.e., values fitting the 1st quartile of PA measured by each instrument, which categorises participants as first quartile/other quartiles) and one ordinal variable “quartiles of physical activity”.

### 2.3. Subjective PA Assessment

To subjectively assess PA, an electronic form of the School Health Action, Planning and Evaluation System (SHAPES) questionnaire has been used [27]. The measurement properties of the SHAPES questionnaire for assessing PA in adolescents are comparable to other recall instruments applied to assess PA in high school children [27]. The questionnaire incorporates two questions requiring a 7-day recall of moderate PA (MPA) and vigorous PA (VPA). MPA was described as “lower intensity physical activities such as walking, riding a bike, and recreational swimming“, whereas VPA was described as “jogging, team sports, fast dancing, jump-rope, and any other physical activity that markedly increased your heart rate and made you breathe hard and sweat“. The participants were instructed to state the number of hours (0–4 h) and minutes in 15-min increments (0–45 min) that MPA and VPA were performed for each day of the previous seven days. For all days at which >4 h of MPA or VPA was reported, the duration of 4:15 h was assumed. Daily duration of MPA and VPA time was calculated as an average response of all 7 days.

### 2.4. Device-Based PA Measurement

The SenseWear Armband^TM^ Pro3 device (BodyMedia Inc., Pittsburgh, PA, USA; SWA) was used to measure the duration and intensity of PA. Accelerometers have long been recognized as feasible, as well as accurate and reliable for objective PA-measurements in large scale studies in children and adolescents [28,29]. In particular, the SWA relies on pattern recognition to estimate EE and the duration and intensity of PA. It uses non-invasive sensors to register heat flux, galvanic skin response, skin temperature, near-body temperature and motion, determined from a biaxial accelerometer. Subsequently, the SWA combines registered data with height, weight, age, gender and handedness into proprietary algorithms to estimate EE and PA duration. It should be acknowledged that the SWA device is no longer supported by the manufacturer. However, the SWA was among few predominately used PA monitors for research purposes over the last decades, and it is still perceived to provide a useful field-criterion of PA assessment [30]. Moreover, a very recent review indicates it is the most accurate among other types of accelerometers [31].

The students wore the SWA device on their right arm, above the m. triceps brachii. Before its activation, each student’s anthropological measures (i.e., height, weight, handedness) and gender were programmed into the device. They were told to wear the SWA for 5 consecutive days (3 schooldays and 2 weekend days) during the entire day and night, excluding time needed for showering or/and other water activities. In order to consider individual results as valid for the analysis, participants had to (i) wear the SWA for a minimum of 10 h of awake time per day and (ii) reach the minimum of 3 days (including at least 1 weekend day) with prescribed wear time, as recommended previously [32]. In addition to wearing the SWA, participants were asked to record their activities, referring to the time spent not wearing the device in a physical activity diary. In accordance with the Compendium of PA for children and youth [33], the data from the diary was added to the SWA outputs. Following the transfer of all sensors’ data averaged over 60-s periods to a computer, valid SWA outputs were analysed. The analysis has been performed using the latest child-specific exercise algorithms v5.2 in SenseWear Professional software v8.1 (BodyMedia Inc., Pittsburgh, PA, USA). The initial release of algorithm v5.0 and its latest update v5.2 have been evaluated in youth versus double labelled water and indirect calorimetry, respectively [34,35]. In comparison to older v2.2, the algorithm v5.2 has showed substantially reduced measurement bias with a special improvement for biking, in both children and adolescents [35,36,37].

A measure used to describe the intensity of PA was the metabolic equivalent (MET). In line with the PA guidelines for Canadians (2002) and the review of Janssen and LeBlanc (2010) [38,39], activities requiring EE between 4 and 7 METs were classified as MPA whereas activities requiring above 7 METs were categorised as VPA. To determine the average of MPA and VPA daily we multiplied the average school day value by 5 and the average weekend day value by 2 and then divided the score by 7, according to the formula: MPA, VPA = ((MEANschooldays ∗ 5) + (MEANweekend days ∗ 2))/7, as described previously [25,26].

### 2.5. Data Analysis

To process the collected data, SPSS software version 24.0 (IBM, New York, NY, USA) was used. For all data analyses, the participants were stratified by sex. Data on descriptive characteristics of the sample, participants’ PA level and level of agreement between subjective and device-based methods for PA assessment are presented as means ± SD and medians (interquartile range) for normally distributed and non-normally distributed continuous variables, respectively, and as percentages for categorical variables. Out of data originally derived from the questionnaire and the SWA device two different types of categorical variables were extracted: dichotomous and ordinal. Dichotomous categorical variables were “sufficient activity”, low activity” and “high activity”. Ordinal categorical variables were quartiles of PA. Differences between questionnaire and SWA in assessing/measuring PA and classifying participants as complying to current PA health recommendations were tested using the paired-sample t-test and uncorrected McNemar’s Chi-square test [40,41], respectively.

Initially, to compare the level of agreement between used questionnaire and device-based methods in ranking adolescents according to the time spent in PA with other self-report methods, Spearman’s rho (ρ) was applied to establish rank-order correlation between the questionnaire and SWA data. Next, to evaluate the ability of the questionnaire to categorise adolescents relative to field-based multiple-sensor monitoring, three different analyses were conducted. Initially, the percent agreement was calculated to present consistency between two measures in categorization within each of the above dichotomous categorical variables. In particular, the analysis of sensitivity and specificity was performed to evaluate the questionnaire’s ability to correctly categorise adolescents as (i) sufficiently or insufficiently active, (ii) the first or the other quartile of PA and (iii) the last or the other quartile of PA versus multiple sensor monitoring as a field-based comparison measure. In this study, specificity refers to the ability of the questionnaire to correctly categorise individuals as sufficiently active/being the first quartile of PA/being the last quartile of PA, whereas the sensitivity indicated the ability to correctly categorise adolescents as insufficiently active/being other than the first quartile/being other than the last quartile of PA in categorical variables “sufficient activity”, low activity” and “high activity”. In addition, specificity was used to show the ability of the questionnaire to correctly allocate adolescents in each of the PA quartiles as determined through device-based measurement. Technically, both questionnaire-based and device-based data were first binary classified and then were compared through the contingency tables 2 × 2. Specificity was calculated as the percentage of individuals classified by both methods as sufficiently active, and sensitivity was computed as the percentage of individuals classified by both methods as inactive. Subsequently, Kappa (κ) and weighted Kappa (κ) statistics were used to evaluate the level of agreement between questionnaire-based and device-based dichotomous variables and ordinal variables, respectively. The following scale has been used to interpret the level of Kappa’s κ: 0–0.20 was rated as “poor”; 0.21–0.40 as “fair”; 0.41–0.60 as “moderate”; 0.61–0.80 as “substantial”; and 0.81–1.0 as “near perfect” [42]. In addition, one-way ANOVA with Bonferroni post-hoc comparison was used to determine differences between questionnaire-based quartiles in duration of PA measured via the multiple-sensor monitor. A post-hoc power analysis was performed to determine if the current sample size provides adequate power to detect correlation and agreement with confidence. The statistical power analysis program G*Power for Windows 3 was used to calculate the power of correlation [43], and a Web-based Sample Size Calculator was applied to evaluate the adequacy of the current sample for the agreement [44]. It revealed that a sample of *n* = 61 boys allows the detection of a medium-to-large effect size (ρ = 0.35 and w = 0.35; [45]) with the power (1–β error probability) of 0.82 and 0.80 for correlation and agreement, respectively, when setting the significance level at *p* = 0.05.

## 3. Results

The descriptions of the study participants (age, body mass index, sum of four skinfolds, time spent in PA of different intensities and compliance with PA recommendations assessed by questionnaire and by SWA device) who were included in the final analysis are displayed in Table 1. Female and male study participants did not differ based on age and BMI. Boys spent significantly more daily time in MPA, VPA and both combined than do the girls, as measured by SWA device. In comparison to girls, a greater portion of boys met the current PA health recommendations of >60 min/day. The recall questionnaire was sensitive enough to detect above significant difference in PA time between boys and girls for VPA only. SHAPES-estimated average time spent in MVPA and VPA was significantly longer than the SWA-measured time in both boys (*p* = 0.005 and *p* < 0.001) and girls (both *p* < 0.001). In contrast, estimated MPA daily time did not differ significantly in girls (*p* = 0.616) and was significantly shorter in comparison to SWA-measured time in boys (*p* = 0.009). A non-parametric McNemar’s Chi-square test detected no meaningful difference (*p* = 0.739) between the device-based method and recall questionnaire in classifying male participants as complying to current PA health recommendations, while significant variation occurred between the two methods in classifying females (*p* = 0.040).

The initial analysis showed a weak correlation between 7-day recall questionnaire and multiple-sensor monitor in ranking male adolescents according to the time spent in MVPA and VPA as presented in Table 2. In contrast, self-reported and field-measured time in three PA-intensities did not demonstrate a substantial level of correlation to individually rank female participants.

The values of the Kappa statistics indicate a high consistency and fair level of agreement between the questionnaire and device-based method in categorising males as complying to the current health recommendations regarding the amount of MVPA, but low consistency and no significant agreement with the device-based method were found in classifying them to the highest or lowest quartile of MVPA (Table 3). Furthermore, the one-way ANOVA detected a substantial difference between questionnaire-based sufficiently active vs. insufficiently active males in relative EE assessed by the SWA (13.6 vs. 9.0 kcal/kg/day; *p* = 0.05). Moderate consistency and no significant agreement were found between the questionnaire and device-based method in categorising females in any of three dichotomous variables and no differences in any of the device-based PA measures were found for the questionnaire-based sufficiently active vs. insufficiently active groups of girls.

Table 4 shows the results of weighted Kappa statistics evaluating the level of agreement between questionnaire-based and device-based allocation of adolescents into the quartiles according to the time spent in different intensities of PA. There was a low consistency and no significant agreement observed between the two methods in categorising both males and females. However, one-way ANOVA with Bonferroni post-hoc comparisons revealed significant differences between the highest and lowest quartiles of MVPA and VPA in males, whereas only trivial differences between quartiles of PA were found in females (Figure 1).

## 4. Discussion

It has been proposed that the PA assessment in public health studies should strive to classify individuals into quantiles according to the amount of PA [20]. This study evaluated the potential of the SHAPES questionnaire to identify insufficiently active adolescents and to rank adolescents into quartiles according to the duration of PA. In accordance with the global and regional trends [46,47,48,49], in the currently studied population, boys had higher levels of PA and tended to meet health guidelines more often than the girls did. The main finding of this study was that the SHAPES might be an appropriate method to discriminate male adolescents at greater health risk due to insufficient PA given the WHO recommendations. Second, the recall questionnaire seems to have reasonable potential to individually rank male adolescents and to distinguish between the most active and the least active quartiles of male adolescents, respectively, according to the duration of MVPA and VPA. In the female group, no ability of the questionnaire was found either to identify compliance to the WHO recommendations, to distinguish among quartiles of PA, or to rank them individually.

Consistent with previous results [14,50], in the present research, adolescents reported notably higher levels of MVPA and VPA in comparison to device-measured values. Because of consistently poor agreement between methods to assess individual amounts of PA [17,51,52], it has been common to use rank-correlation to assess effectiveness of recall questionnaires. In the majority of previously evaluated instruments, the median intra-class correlation coefficients ranged between 0.29 and 0.41 [17,50,53]. The measurement properties of the questionnaire used in the present study have been investigated previously and were in line with other 7-day-recall questionnaires [27]. In male adolescents, a rank-correlation of 0.32 and 0.40 between the SHAPES questionnaire and the SWA data in measuring MVPA and VPA, respectively, suggests that the effectiveness of the used questionnaire seems comparable to results of previously evaluated subjective methods in children and adolescents [17,51].

The practical aim of this study was to examine the agreement of subjective and device-based measures of PA at two different levels: the first level was the ability to distinguish the sufficiently active and insufficiently active adolescents and the most/least active from the rest of the group, respectively, and the second level was to rank adolescents into quartiles based on the level of their habitual PA. The results have demonstrated a high percentage of boys found by both methods as active (i.e., specificity = 92.3%) and a low percentage of boys found by both methods as inactive (i.e., sensitivity = 37.5%). The above led to a high overall percent agreement (85.0%) and fairly significant agreement (κ = 0.32, *p* = 0.014) between the two methods in classifying boys as sufficiently active according to the current PA recommendations. This agreement was supported by the fact that individuals who were classified as sufficiently active by the questionnaire have been found to expend relatively more energy in PA in comparison to inactive individuals, as measured by the SWA device. Conversely, in girls, no satisfactory agreement between self-reported categorization and device-based categorization based on reaching 60 min of MVPA/day was noted, nor was there a difference in device-based relative PAEE between two categories established by the questionnaire. The analysis of agreement between recalls and device-based methods heavily depends on the number of participants that meet or do not meet the recommendations. At the same time, because of inconsistent recall periods and interpretation of PA intensity among questionnaires, as well as different cut-off points, epoch lengths, and recognition patterns in PA measurement among activity monitors [51], large variations in the prevalence of meeting recommendations even among similar populations may occur. Hence, it is not surprising that the present findings disagree with the results observed in two studies that involved Australian adolescents. The authors reported that female and overweight adolescents, identified as less active sub-groups, better concurred in reported PA-level to device-based measurements in comparison to males [18,54]. In our study, boys were identified as more physically active and their self-reporting levels of habitual PA seem to agree better with field-based measurements. It is possible that a portion of the current female population overreported their PA due to subjective sensation of PA levels. A tendency has been observed for overweight girls to overestimate their total amount of PA [55]. Opposite to what was observed in the above study [18], a greater portion of adolescents in the current study has been identified by both methods as meeting the PA recommendations (specificity) than as not meeting the PA recommendations (sensitivity). This is likely because the epoch length in the Australian study was set at high 60 s, and because both SHAPES and the SWA seem to be generally prone to estimate higher levels of time spent in moderate-intensity PA in comparison to equivalent tools [56,57,58,59]. Furthermore, the present study revealed no significant ability of the self-report method to classify adolescents into quartiles of PA as established by multiple sensor accelerometery. Both specificity and overall percent agreement in classifying adolescents into quartiles were notably lower in comparison to preceding dichotomous classification (28.6–92.3 vs. 6.7–46.7% and 60.9–85.0 vs. 21.3–32.8%, respectively). Booth et al. (2002) also found higher values of percent agreement for the two-category (dichotomous) measure compared with the three-category measure of PA [54]. However, male adolescents that were classified by the questionnaire in the first and in the fourth quartile differed on the basis of time spent in MVPA and VPA as measured by multiple-sensor monitor (*p* = 0.032 and *p* = 0.011, respectively). Taken together with the fact that weak rank-correlations have been observed between SWA and SHAPES outcomes for MVPA and VPA and that SHAPES-based assessment concurred with SWA-based measurement in recognizing participants complying to the guidelines, the last result supports the premise that the questionnaire might be able to distinguish only among the most active and least active male adolescents.

The findings of this study should be considered in light of several limitations. The main disadvantage of the current study design is certainly a slight time mismatch between self-report and device-based observation of PA. Still, in order to minimize the effect of this limitation, three different measures were applied: first, self-report and device-based measurement were conducted in a time period no longer than three weeks apart to avoid seasonal discrepancies in regular daily behaviour; second, it was not strictly defined which method would be applied first, and the participants were not informed about the specific methodology of studying the agreement between two methods; and third, after both subjective and device-based observation, the participants were asked to declare if something prevented them from being as active as usual. Only the data of the participants who reported no deviation from the usual routine were included in the analysis. This led to a reduced number of participants taken into analysis and by extension, might have slightly diminished generalizability of the results even though the acceptable power has been reached. Note, however, that the sample was a highly representative group of urban adolescents, and its size remained comparable to previous research [51]. Also, no differences in demographics and questionnaire-reported PA levels were detected between the participants included into analysis and those that dropped out due to missing device-based data or reports on unusual daily routine.

## 5. Conclusions

The present study found that the SHAPES questionnaire may be an effective tool to identify sufficiently active adolescents based on current health recommendations, and to distinguish between the most active and the least active quartiles of male adolescents. On the other hand, it supports previous findings that subjective methods overestimate the amount of PA in adolescents, which most likely leads to insufficient sensitivity of the questionnaire to rank adolescents into quartiles based on device-measured time spent in PA, as concluded in the current results preliminary presentation [60]. Future studies should strive to simultaneously apply subjective and device-based methods in a larger sample of adolescents in order to confirm the results of this study.

## Figures and Tables

**Figure 1 sensors-21-01986-f001:**
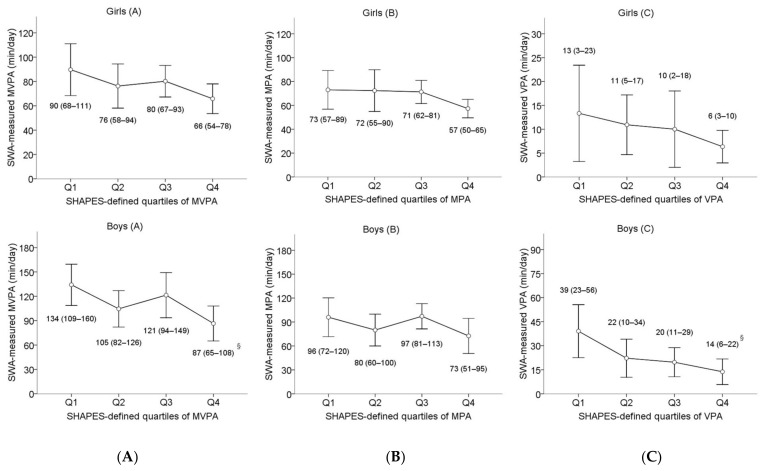
Differences between questionnaire-based quartiles in objectively measured time spent at MVPA (**A**), MPA (**B**) and VPA (**C**) for female and male adolescents determined using one-way ANOVA. Data are reported as Mean (95% CI). Note: MVPA = moderate-to-vigorous physical activity; MPA = moderate physical activity; VPA = vigorous physical activity; ^§^ differences were significant between the 1st and the 4th quartiles (MVPA *p* = 0.032; VPA *p* = 0.011)

**Table 1 sensors-21-01986-t001:** Basic descriptive parameters of the sample and differences between female and male study participants determined using t-test and Mann-Whitney U test.

	Boys (*n* = 61)	Girls (*n* = 89)	*p*
Age (yrs)	15.6 ± 0.4	15.6 ± 0.3	0.921
BMI (kg/m^2^)	20.9 ± 3.0	20.9 ± 2.8	0.991
S4SF (mm)	33.5 ± 13.1	47.5 ± 13.8	0.000
SHAPES questionnaire			
MVPA (min/day)	140.1 ± 78.2 **	118.7 ± 72.1 **	0.092
MPA (min/day)	67.7 ± 51.8 **	65.4 ± 51.2	0.792
VPA (min/day)	72.4 ± 40.4 **	53.3 ± 41.2 **	0.005
Meeting PA recommendations (%)	88.5	80.9 *	0.212
SWA device			
MVPA (min/day)	111.9 ± 46.6	77.8 ± 36.6	0.000
MPA (min/day)	87.3 ± 37.9	68.5 ± 29.4	0.001
VPA (min/day)	23.6 ± 22.8	10.1 ± 16.6	0.000
Meeting PA recommendations (%)	86.7	66.7	0.006

Note: BMI = body mass index; S4SF = sum four skinfolds; MVPA = moderate-to-vigorous physical activity; MPA = moderate physical activity; VPA = vigorous physical activity; *p* = the level of statistical significance for difference between females and males; ** statistically different at level *p* < 0.01 from the corresponding PA-intensity time measured using SWA-device; * statistically different at level *p* < 0.05 from the portion established using SWA-device.

**Table 2 sensors-21-01986-t002:** Spearman’s rank correlations between the questionnaire-based assessment and SWA-based measures of the time spent in MVPA, MPA and VPA for female and male adolescents.

	Boys		Girls	
	ρ	*p*	ρ	*p*
MVPA (min/day)	0.323	0.012	0.126	0.247
MPA (min/day)	0.136	0.296	0.149	0.164
VPA (min/day)	0.396	0.002	0.125	0.243

Note: MVPA = moderate-to-vigorous physical activity; MPA = moderate physical activity; VPA = vigorous physical activity; ρ = Spearman’s coefficient of rank correlations; *p* = the level of statistical significance.

**Table 3 sensors-21-01986-t003:** Agreement between the questionnaire-based and SWA-based categorization of female and male adolescents as sufficiently active based on current health recommendations, as well as the most active quartile and the least active quartile according to MVPA determined using percent agreement and Kappa statistics.

	Specificity (%)	Sensitivity (%)	Agreement (%)	κ	*p*
Boys					
Sufficient activity (60 min MVPA/day)	92.3	37.5	85.0	0.315	0.014
Low activity (4th quartile of MVPA)	40.0	82.2	71.7	0.227	0.078
High activity (1st quartile of MVPA)	40.0	80.0	70.0	0.200	0.121
Girls					
Sufficient activity (60 min MVPA/day)	81.0	20.7	60.9	0.019	0.848
Low activity (4th quartile of MVPA)	31.8	76.9	65.5	0.087	0.415
High activity (1st quartile of MVPA)	28.6	78.8	66.7	0.075	0.485

Note: BMI = body mass index; S4SF = sum four skinfolds; MVPA = moderate-to-vigorous physical activity; MPA = moderate physical activity; VPA = vigorous physical activity; κ = Kappa agreement; *p* = the level of statistical significance.

**Table 4 sensors-21-01986-t004:** Agreement between the questionnaire-based and SWA-based categorization of female and male adolescents into the quartiles according to the time spent at different intensities of PA determined using percent agreement and Kappa statistics.

	Specificity (%)				Agreement (%)	κ	*p*
	Q1	Q2	Q3	Q4			
Boys	40.0	20.0	6.7	40.0	26.7	0.022	0.766
MVPA quartiles	13.2	20.0	37.5	46.7	29.5	0.059	0.427
MPA quartiles	46.7	20.0	31.3	33.3	32.8	0.103	0.163
VPA quartiles							
Girls	28.6	22.7	27.3	31.8	27.6	0.034	0.581
MVPA quartiles	27.3	22.7	26.1	22.7	24.7	−0.004	0.943
MPA quartiles	27.3	27.3	26.1	31.8	21.3	0.041	0.502
VPA quartiles	40.0	20.0	6.7	40.0	26.7	0.022	0.766

Note: Q = quartile; MVPA = moderate-to-vigorous physical activity; MPA = moderate physical activity; VPA = vigorous physical activity; κ = Kappa agreement; *p* = the level of statistical significance.

## Data Availability

The raw data used to support the findings of this study are available from M.S. upon reasonable request.

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
