# Peer review of "Agreement between the SHAPES Questionnaire and a Multiple-Sensor Monitor in Assessing Physical Activity of Adolescents Using Categorial Approach: A Cross-Sectional Study"

_sensors, 2021, doi:10.3390/s21061986_

Round 1

Reviewer 1 Report

From 47 to 49

In order to control physical activity among adolescents, wide range of devices such accelerometers, pedometers, multiple-sensor PA monitors, fitness trackers and smartphone technologies have been recommended as valid and accurate [7–10].

From 133 to 136

However, the SWA was among few predominately used PA-monitors in research purpose over the last decades, and it is still perceived to provide a useful field-criterion of PA assessment [24]. Moreover, a very recent review indicates it is the most accurate among other types of accelerometers [25].

The author explained why he chose SWA among accelerometers but did not explained why accelerometer among the other devices.

From 139 to 143

They were told to wear the SWA for 5 consecutive days (3 schooldays and 2 weekend days) during the entire day and night, excluding time needed for showering or/and other 4 of 12  water activities. In order to consider individual results as valid for the analysis, participants had to (i) wear the SWA for a minimum of 10 h of awake time per day and (ii) reach the minimum of 3 days (including at least 1 weekend day) with prescribed wear time, as recommended previously.

Although the authors lowered the criteria for inclusion in the study (5 to 3 days; 24 to 10 hours), only half of the respondents completed the study.

This needs a comment

Activities  requiring energy expenditure between 4 and 7 METs were classified as MPA whereas activities  requiring above 7 METs were categorized as VPA.

The authors need to explain how they determined this?

Subjective measurement 7 days, objective measurement 3 days. In order to be able to compare the results, the authors calculated daily averages (working days and weekends).

Iz this comon method or the authors created such a method themselves.

The authors had a very similar abstract at a congress in Italy (probably from the same project), the abstract is entitled: Agreement between Pa-Questionnaire and Multiple Sensor Monitor in Categorizing Physical Activity in Male Adolescents

From the conclusion of this abstract:

It supports previous findings that subjective methods overreport the amount of PA in adolescents, which most likely leads to insufficient sensitivity of the questionnaire to rank adolescents into quartiles based on objective time spent in PA.

Maybe this abstract should be quoted in the article?

Reviewer 2 Report

There is a substantial literature base on reliability / validity of PA questionnaires, and the value of additional studies such as this is questionable. I suggest the authors provide more context to the assertion in line 70 that such studies are scarce. Maybe start with a Google scholar search with the terms "physical activity reliability questionnaire adolescents review", and a brief summary of how all these studies have failed to meet the criteria of "...reporting the level of agreement between questionnaires and PA monitors for categorizing individuals as physically active or ranking them as low/highly active". It's entirely possible that none of these studies categorised physical activity data as the authors of this study have, but that seems unlikely. 

In general, the article could use some additional review from a native english speaker. There are many unusually written parts in this paper, but overall readers would be able to make sense of it. A minor example is the many situations in which "a" or "the" has not been included in the sentence when it should be. Another example is line 77 "longitudinal study shaped to investigate the health-related habits".... why is the study 'shaped'?. There are many other such examples, and convoluted sentence structure that could be improved. 

Use of wilcoxon sign rank tests seem to be inappropriate. The way the authors have described the analysis, they have used wilcoxon as a test of association, testing for classification accuracy. Chi-square tests in this instance would be correct. 

Table 1 seems inconsistent in the use of asterisks to represent significant p values. 

Overall, if the journal sees this article as suitable and the authors wish to put their names to this, I see no reason to object. 

Round 2

Reviewer 1 Report

All  all necessary changes have been made, the authors listed all the limitations of the study. 

Author Response

Dear reviewer,

The authors appreciate your positive opinion regarding our manuscript and would like to thank you for all your helpful comments.